# Breast cancer patient stratification using domain adaptation based lymphocyte detection in HER2 stained tissue sections

**Ansh Kapil**[1]                                                                 ANSH.KAPIL@ASTRAZENECA.COM
**Armin Meier**[1]                                                              ARMIN.MEIER@ASTRAZENECA.COM
**Anatoliy Shumilov**[1]                                         ANATOLIY.SHUMILOV@ASTRAZENECA.COM
**Susanne Haneder**[1]                                          SUSANNE.HANEDER@ASTRAZENECA.COM
**Helen Angell**[2]                                                         HELEN.ANGELL@ASTRAZENECA.COM
**Günter Schmidt**[1]                                            GUENTER.SCHMIDT@ASTRAZENECA.COM

[1] *AstraZeneca Computational Pathology, Oncology R&D, Munich, Germany*

[2] *AstraZeneca Translational Medicine, Oncology R&D, Cambridge, UK*

**Editors:** Under Review for MIDL 2021

## Abstract

We extend the CycleGAN architecture with a style-based generator and show the efficacy of the proposed domain adaptation-based method between two histopathology image domains - Hematoxylin and Eosin (H&E) and HER2 immunohistochemically (IHC) images. Using the proposed method, we re-used large set of pre-existing annotations for detection of tumor infiltrating lymphocytes (TILs), which were originally done on H&E, towards a TIL detector applicable on HER2 IHC images. We provide analytical validation of the resulting TIL detector. Furthermore, we show that the detected stromal TIL densities are significantly prognostic as a biomarker for patient stratification on a triple-negative breast cancer (TNBC) cohort.

**Keywords:** Computational Pathology, Deep Learning, Domain Adaptation, Breast Cancer

## 1. Introduction

Tumor infiltrating lymphocytes (TILs) are indicators of improved prognosis in breast cancer. TILs can be characterized as round to polygonal relatively small cells with little cytoplasm and a nucleus with homogeneous texture. In this work, we build a deep learning based system to detect TILs on HER2 stained IHC images. We aim to minimize the effort needed to collect large amounts of ground truth annotations for TIL detection by using a domain adaptation-based system that will re-use pre-existing annotations on H&E slides from an independent cohort to train a model that works on IHC (HER2 stained) images. We show clinical relevance of this method by performing patient stratification on an unseen cohort with n=145 samples. Patient stratification is the task of identifying a sub-cohort of patients with improved overall or progression-free survival; in this case by using stromal TIL (sTIL) density as a biomarker.

## 2. CycleGAN with Adaptive Instance Normalization

We augment the original CycleGAN architecture (Zhu et al., 2017) by a style encoder to make it adaptive to any given input style using Adaptive Instance Normalization (AdaIN).

$$AdaIN(x, y) = \sigma(y) \left( \frac{x - \mu(x)}{\sigma(x)} \right) + \mu(y)$$

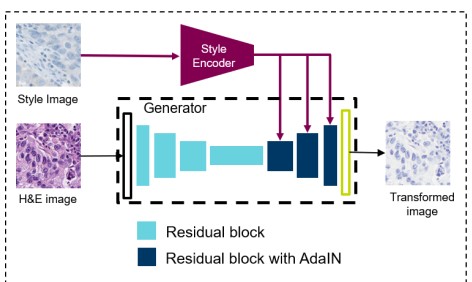 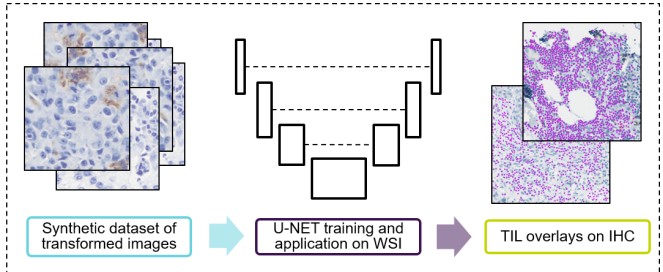

Figure 1: Left: Modified CycleGAN generator with AdaIN. Style encoder E is used to learn style features from a style image. The full CycleGAN architecture is not shown for clarity purposes. Right: A UNet model with EfficientNet backbone was trained on synthetic IHC images only using the annotations transferred from corresponding H&E patches.

The input content x is first instance normalized (IN) (Ulyanov et al., 2016) and then scaled and shifted with parameters obtained from style features y. The style features are computed from an image encoded vector from the style image. To incorporate this, the generator network of the CycleGAN needs an additional input - the style vector, which is computed through another CNN - the style encoder $E$. The schema of the modified CycleGAN generator is shown in Figure 1. With the incorporation of style encoders $E_A$ and $E_B$, the two adversarial losses $\mathcal{L}_{GAN}^{AB}$ and $\mathcal{L}_{GAN}^{BA}$ are modified, while other losses like the cycle loss remain the same. Let $x_A$, $x_B$ represent the content images and $y_B$, $y_A$ represents the respective style image in domains A and B. Modified losses are as follows:

$$\min_{G_{AB}} \max_{D_B} \mathcal{L}_{GAN}^{AB} := \mathbb{E}_{x_B \sim \mathcal{X}_B} \log(D_B(x_B)) + \mathbb{E}_{x_A \sim \mathcal{X}_A, y_B \sim \mathcal{X}_B} \log(1 - D_B(G_{AB}(x_A, E_B(y_B))))$$

$$\min_{G_{BA}} \max_{D_A} \mathcal{L}_{GAN}^{BA} := \mathbb{E}_{x_A \sim \mathcal{X}_A} \log(D_A(x_A)) + \mathbb{E}_{x_B \sim \mathcal{X}_B, y_A \sim \mathcal{X}_A} \log(1 - D_A(G_{BA}(x_B, E_A(y_A))))$$

The development set consisted of 60k H&E patches with TILs annotations from an independent cohort. The aforementioned Adaptive CycleGAN model was used to generate synthetic IHC patches from H&E patches. Once the patches were synthesized, we were able to re-use exactly the same ground truth masks as H&E domain for model training to train a UNet with EfficientNet backbone. The model was applied to the whole slide image (WSI) in a sliding window manner and cell centers were obtained from final posterior maps using thresholding followed by non-maximal suppression.

## 3. Results

Analytical validation of the TIL detector was done against pathologist's TIL annotations performed on HER2 IHC images. N=38 regions of interests (ROI) were annotated by one pathologist that contained 3122 TIL annotations for testing. Detected TIL locations were compared against the annotated TIL locations using Hungarian matching and an F1-score of 0.68 was obtained. The absolute TIL count correlation between amount of detected and annotated TILs (both on log scale) showed spearman correlation 0.93. For clinical validation, the model was applied on a TNBC patient cohort of N=145 patients. The

patients were treated with standard of care treatment (surgery and chemo/radiotherapy) and information on Overall Survival (OS) and Progression-free Survival (PFS) was analyzed. TIL density in tumor associated stromal regions (sTILs density) was computed within a pathologist delineated tumor core region leaving out areas of DCIS and normal tissue. An in-house epithelial detection (two clas model segmenting epithelial regions from others) model was used to detect and exclude the epithelial regions within the tumor core region to obtain area for TIL density computation in stromal regions (number of TILs/$mm^2$). We performed a 2-fold pre-validation to determine a sTIL density cut-point to stratify the patients with respect to OS and PFS into two groups. Log-rank test was performed to find out the significance of these two groups. We observed that the sTIL density is significantly prognostic for OS (log-rank p-value = 0.0062, HR=0.454) as well as for PFS (log-rank p-value = 0.0033, HR=0.463).

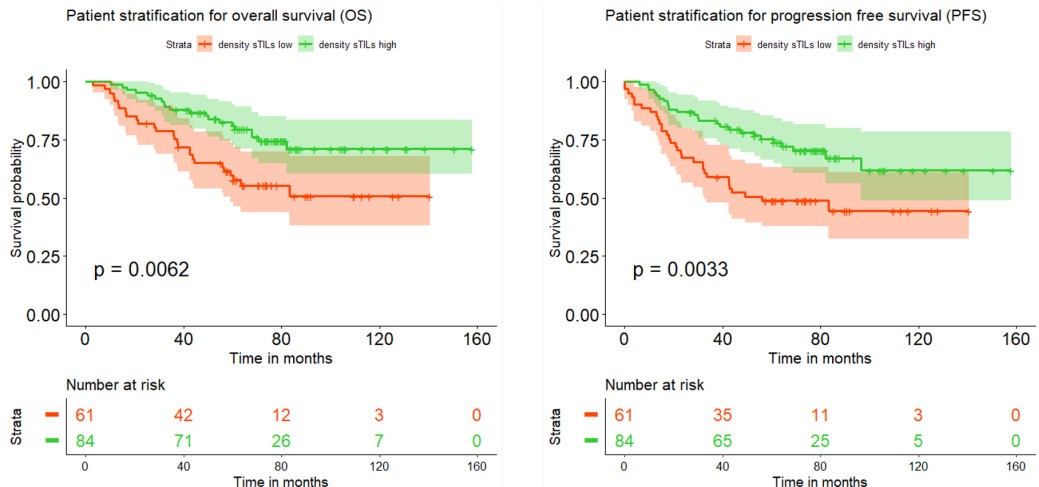

Figure 2: Kaplan-Meier plot for patient stratification with OS and PFS.

## 4. Discussion and outlook

We propose a novel workflow to re-use annotations effectively between different domains. The TIL detector obtained provides analytically and prognostically relevant TIL statistics. This approach will enable us in the future to reuse large sets with pre-existing annotations on multiple image domains to train effective models and obtain novel biomarkers.

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
