# OpenReview forum: "Breast cancer patient stratification using domain adaptation based lymphocyte detection in HER2 stained tissue sections"
_MIDL.io/2021/Conference/Short — MIDL 2021 Poster_

### Official Review · Reviewer_5GRW · 2021-04-29

**Confidence:** 5
**Final Rating:** 3

**Summary:**

The authors develop a workflow which uses pre-existing TIL annotations on H&E to locate TILs on IHC. First, H&E images are synthetically converted to IHC using style-transfer. Next, a U-Net is trained using the synthetic IHC images (with original H&E TIL annotations) to detect TILs in real IHC. Finally, the workflow is tested by using TIL statistics as covariates in a survival model

**Strengths:**

This paper shows an interesting and important use-case for reusing pre-existing annotations which can be expensive and time-consuming to acquire. This project also shows successful domain transfer by validating a TIL detection on survival prognosis by a model trained on synthetically generated images.

Clear diagram

**Weaknesses:**

Although the use-case is important and the workflow unique, no novel methodology is contributed

method is not compared to anything else. Is the survival task easy or difficult?

minor: red-green colorblindness is the most common form of colorblindness. kaplan-meier plots can be improved with a different color combination

**Deanonymize Review:**

no

**Detailed Comments:**

Survival prognosis does not validate accuracy of TIL detection, but rather, the accuracy of general TIL density.  F1 score was rather low which means that the model is not necessarily finding exact TIL locations. However, the good survival prognosis shows that a general count of TILs is enough for this use-case. Style transfer doesn't guarantee accuracy in TIL detection. This is okay as the authors only claim that the model provide clinically and prognostically relevant TIL statistics.



**Justification Of The Rating:**

authors provide an interesting application and conceptual workflow for an important task, as obtaining high-level annotations is difficult. validation of TIL detection after style transfer was not encouraging. however, for this application it was enough to produce a significant survival model.

**Paper Type:**

validation/application paper

**Special Issue:**

no

---

### Official Review · Reviewer_SACD · 2021-04-30

**Confidence:** 4
**Final Rating:** 3

**Summary:**

In this paper, the authors train a cycleGAN to convert a data set of H&E images, for which annotations for TILs exist, into artificial HER2 stained images. They afterwards use the HER2 images and the annotations to train a TIL detector for HER2.
The TIL detector is evaluated against TIL annotations made by a pathologist using real HER2 images. The authors report an F1 score of 0.68 and a spearman correlation of the number of TILs of 0.93. In a further experiment, the detector was used to stratify a partient cohort into 2 subgroups of different overall survival.

**Strengths:**

* The paper shows a good application of style-transfer to re-use precious annotations that are otherwise not applicable to the problem
* The paper is well written and easy to follow
* The paper follows general scientific principles

**Weaknesses:**

* F1 score of is quite low, while the spearman correlation is quite high. These values should be discussed and put in perspective
* The explanation experiment for patient stratification is very brief considering the many information needed to be given (manual annotations and in-house classifier involved)
* The authors should have stated whether such stratification based on TIL density is new or already known. In case of the latter, the appropriate reference is missing

**Deanonymize Review:**

no

**Justification Of The Rating:**

The paper shows a good application of cylceGANs for style transfer in order to spare annotations or to re-use otherwise useless annotations. However, the stratification experiment is described too briefly and the results should have been discussed more.

**Paper Type:**

methodological development

**Special Issue:**

no

---

### Meta-Review · Program_Chairs · 2021-05-06

**Recommendation:** Accept (Poster)
**Confidence:** 5

**Metareview:**

This paper is a clear acceptance. Authors are suggested to address reviewer suggestions in final version.

---

### Decision · Program_Chairs · 2021-05-11

Accept (Poster)